# Carbon Dioxide and Methane Fluxes at the Air-Sea Interface of Red Sea Mangroves

**Mallory A. Sea[1], Neus Garcias-Bonet[1], Vincent Saderne[1]\* and Carlos M. Duarte[1]**

[1] {King Abdullah University of Science and Technology (KAUST), Red Sea Research Center (RSRC), Thuwal, 23955-6900, Saudi Arabia}

*Correspondence to: V. Saderne (vincent.saderne@kaust.edu.sa)

**Abstract**

Mangrove forests are highly productive tropical and subtropical coastal systems that provide a variety of ecosystem services, including the sequestration of carbon. While mangroves are reported to be the most intense carbon sinks among all forests, they can also support large emissions of greenhouse gases (GHG), such as carbon dioxide ($CO_2$) and methane ($CH_4$), to the atmosphere. However, data derived from arid mangrove systems like the Red Sea are lacking. Here, we report net emission rates of $CO_2$ and $CH_4$ from mangroves along the eastern coast of the Red Sea, and assess the relative role of these two gases in supporting total GHG emissions to the atmosphere. Diel $CO_2$ and $CH_4$ emission rates ranged from -3452 to 7500 µmol $CO_2$ $m^{-2}$ $d^{-1}$ and from 0.9 to 13.3 µmol $CH_4$ $m^{-2}$ $d^{-1}$, respectively. The rates reported here fall within previously reported ranges for both $CO_2$ and $CH_4$, but maximum $CO_2$ and $CH_4$ flux rates in the Red Sea are 10 to 100-fold below those previously reported for mangroves elsewhere. Based on the isotopic composition of the $CO_2$ and $CH_4$ produced, we identified potential origins of the organic matter that support GHG emissions. In all but one mangrove stand, GHG emissions appear to be supported by organic matter from mixed sources, potentially reducing $CO_2$ fluxes and instead enhancing $CH_4$ production, a finding that highlights the importance of determining the origin of organic matter in GHG emissions. Methane was the main source of $CO_2$-equivalents despite the comparatively low emission rates in most of the sampled mangroves, and therefore

deserves careful monitoring in this region. By further resolving GHG fluxes in arid mangroves,
we will better ascertain the role of these forests in global carbon budgets.


## 1    Introduction

Mangrove forests, typically growing in the intertidal zones of tropical and subtropical coasts, are highly productive components of coastal ecosystems and adapted to high salinity and anoxic conditions associated with waterlogged sediments. Mangrove forests cover a global estimated area of 137,760 $km^2$ (Giri et al., 2011) and are typically constrained by temperature, with greatest biomass and species diversity in the equatorial zone (Alongi, 2012). Mangroves rank amongst the most threatened ecosystems in the biosphere, with losses estimated at 50% of their global extent over the past 50 years (Alongi, 2012). These losses affect nearly all mangrove regions but the Red Sea, where mangrove coverage has increased by 12% over the past four decades (Almahasheer et al., 2016).

Loss of mangrove forest represents a loss of valuable ecosystem services, including habitat and nursery for marine species, coastal protection from erosion due to wave action, and the filtration of harmful pollutants from terrestrial sources (Alongi, 2008), as well as loss of $CO_2$ sink capacity. Additionally, mangroves can become a source of greenhouse gas (GHG) emissions from disturbed soil carbon stocks (Donato et al., 2011; Alongi, 2014). Hence, mangrove conservation and restoration have been proposed as important components of so-called Blue Carbon strategies to mitigate climate change (Duarte, et al., 2013). Indeed, mangroves are reported to be the most intense carbon sinks among all forests, supporting carbon sequestration rates and organic carbon stocks as much as five times higher than those in terrestrial forests (Donato et al., 2011). While mangrove forests cover less than 1% of total coastal ocean area, they contribute to almost 15% of total carbon sequestration in coastal ecosystems (Alongi, 2012), making mangrove forests highly effective in terms of carbon sequestration per unit area. The management of mangroves to maximize $CO_2$ removal and subsequent storage is gaining momentum as a cost-effective strategy to mitigate climate change.

However, mangrove forests act as both carbon sinks and sources and have been reported to support large GHG emissions in the forms of $CO_2$ and $CH_4$ (Allen et al., 2007; Kristensen et al., 2008a; Chen et al., 2016). Whereas concerns are focused on GHG emissions following mangrove disturbance, estimated at $0.02 - 0.12$ Pg C $yr^{-1}$ globally (Donato et al., 2011), undisturbed

mangrove sediments also support GHG emissions (Purvaja and Ramesh, 2000; Kristensen et al.,
2008b; Chauhan et al., 2015). Recent reports specifically highlight the importance of methane in
flux estimates, as emissions of $CH_4$, with a higher global warming potential, can offset mangrove
carbon burial by as much as 20% (Rosentreter et al., 2018b).  Previous studies on GHG emission
rates either focus on the soil-atmosphere interface, highlighting substantial flux ranges with
mangroves reported to act as negligible (Alongi, 2005) to considerable sources (Livesley and
Andrusiak, 2012; Chen et al., 2016), or examine net fluxes at the air-sea interface, with few
studies in arid systems. Comparisons of carbon sequestration rates between mangrove stands
have revealed that climatic conditions play an important role, with mangroves in the arid tropics,
such as those in the Red Sea, supporting the lowest carbon sequestration rates (Almahasheer et
al. 2017). Likewise, GHG emissions from mangrove forests may vary with climate, with most
reported rates to-date derived from the wet tropics (Alongi et al., 2005; Chauhan et al., 2015;
Chen et al., 2016). Whereas Red Sea mangroves are considered to play a minor role as $CO_2$
sinks, their role may be greater than portrayed by low carbon burial rates if they also support
very low GHG emissions, thereby leading to a balance comparable to mangroves in the wet
tropics.
Here we report air-sea emission rates of $CO_2$ and $CH_4$, along with their carbon isotopic
composition, from incubations of inundated mangrove sediments cores along the Saudi coast of
the Red Sea. We assess the relative role of these two gases in supporting total GHG emissions as
well as their fluctuations along the day-night cycle.

**2      Materials and Methods**

**2.1    Study area**

We sampled seven mangrove forests along the eastern coast of the Red Sea (Fig. 1). We
collected triplicate sediment cores by inserting translucent PVC tubes (30.5 cm in height and 9.5
cm in diameter) into mangrove sediments, collecting approx. 20 cm of sediment and a top
seawater layer. The overlying water was regularly replaced by fresh seawater from the
corresponding station in order to fill the remaining core volume and to measure $CO_2$ and $CH_4$
fluxes from underlying sediments during incubations. Mangrove sediments were sampled five to

ten meters from the forest edge, typically in the center of the mangrove belt. We sampled two stations (S1 and S2) in January and February 2017 and the other five mangrove stations (S3-S7) in March on board the R/V Thuwal as part of a scientific cruise. The cores collected from S1 and S2 were immediately transported to the laboratory, placed in seawater baths and enclosed in environmental growth chambers (Percival Scientific Inc., Perry, IA, USA) with 12:12 light cycles at a constant temperature of 26°C. The sediment cores collected during the scientific cruise were transported immediately on board and placed in open aquarium tanks with running seawater in order to keep them close to *in situ* temperature. Salinity and temperature were routinely recorded using a CTD. Additionally, sediment chlorophyll *a* and nutrient (organic carbon and nitrogen) content were analyzed from cores collected during the scientific cruise.

## 2.2    Sediment characteristics

The chlorophyll *a* content of the sediment was measured by fluorometry. The surface layer of each replicate core was collected and frozen until further analysis. Prior to chlorophyll *a* extraction, the sediment samples were left at room temperature to thaw. The chlorophyll *a* was extracted by adding 7 ml of 90% acetone to 2 ml of sediment sample. After a 24h incubation at $4^o$ C in dark conditions, the samples were centrifuged and the chlorophyll *a* content in the supernatant was measured on a Trilogy fluorometer. The nutrient (organic carbon and nitrogen) content of the sediment was analyzed on an Organic Elemental Analyzer (Flash 2000) after acidification of sediment samples.——

## 2.3    Measurement of greenhouse gas fluxes

We measured $CO_2$ and $CH_4$ air-sea fluxes using two different techniques. The $CO_2$ and $CH_4$ fluxes from stations S1 and S2 were measured using the closed water circuit technique and the $CO_2$ and $CH_4$ fluxes from the rest of the stations sampled during the scientific cruise (S3-S7) were measured using the headspace technique.

### 2.3.1  Measurement of $CO_2$ and $CH_4$ fluxes in sediment core incubations using closed water circuit technique


We incubated mangrove sediment cores from stations S1 and S2 using a closed water circuit
technique in order to measure changes in $CO_2$ and $CH_4$ concentrations. Before starting the
incubation, the seawater above the sediment from each core was replaced by fresh seawater
collected from the same location, avoiding disturbance of the sediment. Then, the seawater from
the core was recirculated by a peristaltic pump in an enclosed water circuit through a membrane
equilibrator (Liqui-cel mini module, 3M, Minnesota, USA). This setup enables the equilibration
of gases in dissolution with an enclosed air circuit. The air from the enclosed air circuit was then
passed through a desiccant column (calcium sulfate, WA Hammond Drierite Co., LTD, Ohio,
USA) and flowed into a cavity ring-down spectrometer (CRDS; Picarro Inc., Santa Clara, CA,
USA) to continuously measure the $CO_2$ and $CH_4$ concentration. We ran the incubations for at
least 30 minutes under light (200 µmol photons $m^{-2}$ $s^{-1}$) and dark conditions.

The concentration of $CO_2$ in the water circuit (µmol $ml^{-1}$) was calculated by Eq. (1):
$[CO_2] = Hcp \times [HP\_CO_2] \times (1 - pH_2O)$,                      (1)
where Hcp is the Henry constant (mol $ml^{-1}$ $atm^{-1}$) calculated using R marelac package (Soetaert
et al., 2016); $[HP\_CO_2]$ is the given concentration of $CO_2$ (ppm), and $pH_2O$ is the water vapor
pressure (atm).
The $CO_2$ fluxes were calculated from the change in $CO_2$ concentration over time during our
incubations, correcting by the seawater volume present in each core. Then, the fluxes were
transformed to an aerial basis (µmol $m^{-2}$ $h^{-1}$) by taking into account the core surface area. Finally,
the daily fluxes (µmol $m^{-2}$ $d^{-1}$) were calculated by multiplying the $CO_2$ flux obtained under light
conditions by the number of light hours plus the $CO_2$ flux obtained under dark conditions by the
number of dark hours.
The $CH_4$ fluxes were calculated in the same manner as for the $CO_2$ fluxes, with the exception
that the Henry constant was calculated using Eq. (2):
$\beta = Hcp \times (RT)$,                            (2)
where Hcp is the Henry constant (mol ml$^{-1}$ atm$^{-1}$), $R$ is the ideal gas constant (82.057338 atm ml
mol$^{-1}$ K$^{-1}$), $T$ is standard temperature (273.15 K), and β is the Bunsen solubility coefficient of
CH$_4$, extracted from Wiesenburg and Guinasso (1979).

## 2.3.2 Measurement of CO$_2$ and CH$_4$ fluxes in sediment core incubations using the headspace technique

Mangrove sediment cores from stations S3 to S7 were incubated using a headspace technique in
order to measure changes in CO$_2$ and CH$_4$ concentrations. Before starting the incubation, the
seawater above the sediment from each core was replaced by fresh seawater from the running
seawater system, leaving a headspace of 200 ml. Each core was sealed with a stopper equipped
with a gas-tight valve serving as a headspace sampling port. The sealed core was left for 1 hour
before the first headspace sampling to allow equilibration between seawater and air phases. Each
core was sampled with a syringe, withdrawing 15 ml of air from the equilibrated headspace.
Headspace samples were periodically drawn from each sediment incubation over a 24-hour
incubation period. The CO$_2$ and CH$_4$ concentrations in the headspace samples along with their
isotopic composition ($\delta^{13}$C-CO$_2$ and $\delta^{13}$C-CH$_4$) were measured with a CRDS (Picarro Inc., Santa
Clara, CA, USA) connected to a small sample isotopic module extension (SSIM A0314, Picarro
Inc., Santa Clara, CA, USA). We ran standards (730 ppm CO$_2$, 1.9 ppm CH$_4$) before and after
every three samples.
The concentration of dissolved CO$_2$ in the seawater after equilibrium was calculated from the
concentration in the equilibrated headspace (ppm) as described previously by Wilson et al.
(2012) for other gases:
$[CO_2]_w = 10^{-6}\ \beta\ m_a\ p_{dry}$,  (3)
where β is the Bunsen solubility coefficient of CO$_2$ (mol ml$^{-1}$ atm$^{-1}$), $m_a$ is the given
concentration of CO$_2$ in the equilibrated headspace (ppm), and $p_{dry}$ is atmospheric pressure (atm)
of dry air. The Bunsen solubility coefficient of CO$_2$ was calculated using Eq. (4):
$\beta = \text{Hcp x } (RT)$  (4)
where Hcp is the Henry constant (mol ml$^{-1}$ atm$^{-1}$) calculated using R marelac package (Soetaert
et al., 2016), $R$ is the ideal gas constant (82.057338 atm ml mol$^{-1}$ K$^{-1}$) and $T$ is standard
temperature (273.15 K). The atmospheric pressure of dry air ($p_{dry}$) was calculated using Eq. (5):
$p_{dry} = p_{wet} (1 - \%H_2O)$           (5)
where $p_{wet}$ is the atmospheric pressure of wet air corrected by the effect of multiple syringe
draws from the same core, applying Boyle's law.
The initial concentration of dissolved $CO_2$ in seawater before equilibrium was then calculated as:
$[CO_2]_{aq} = ([CO_2]_w V_w + 10^{-6} m_a V_a) / V_w$         (6)
where $[CO_2]_w$ is the concentration of dissolved $CO_2$ in the seawater after equilibrium, $V_w$ is the
volume of seawater (ml) and $V_a$ is the headspace volume (ml) in the core. Finally, treating the
gas as ideal, the units were converted to nM using Eq. (7):
$[CO_2]_{aq} = 10^9 * p_{dry}[CO_2]_{aq} / (RT)$         (7)
where $R$ is the ideal gas constant (0.08206 atm l mol$^{-1}$ K$^{-1}$) and $T$ is temperature (K).
The $CO_2$ fluxes were calculated from the change in $CO_2$ concentration over time during our
incubations, correcting by the seawater volume present in each core. Then, the fluxes were
transformed to an aerial basis ($\mu mol\ m^{-2}\ d^{-1}$) by taking into account the core surface area. Finally,
the day and night fluxes ($\mu mol\ m^{-2}\ h^{-1}$) were calculated from the change in $CO_2$ concentration
between consecutive samplings during day and night time, respectively.
The $CH_4$ fluxes were calculated in the same manner as for the $CO_2$ fluxes, with the exception
that the Bunsen solubility coefficient of $CH_4$ was calculated according to Wiesenburg and
Guinasso (1979).

**2.4    Isotopic composition of $CO_2$ ($\delta^{13}C$- $CO_2$) and $CH_4$ ($\delta^{13}C$- $CH_4$)**

The isotopic signature of the $CO_2$ and $CH_4$ produced during incubations was estimated by
conducting keeling plots (Pataki et al. 2003; Thom et la. 2003; Garcias-Bonet and Duarte 2017).
Briefly, the $\delta^{13}C$ of the $CO_2$ and $CH_4$ produced was extracted from the intercept of the linear
regression between the inverse of the gas partial pressure and the isotopic signature.

The data set is available from Sea et al. (2018).

**3      Results**

The mean ($\pm$ SE) diel $CO_2$ and $CH_4$ emission rates for the seven sites were $372 \pm 1309$ µmol $CO_2$
$m^{-2}$ $d^{-1}$ and $5.6 \pm 1.6$ µmol $CH_4$ $m^{-2}$ $d^{-1}$, respectively. We observed high variability among the
seven mangrove forest sites studied, with net $CO_2$ and $CH_4$ diel emission rates ranging from
-3452 to 7500 µmol $CO_2$ $m^{-2}$ $d^{-1}$ and from 0.9 to 13.3 µmol $CH_4$ $m^{-2}$ $d^{-1}$, respectively (Table 1).

Mangrove sediments absorbed $CO_2$ during daytime and emitted $CO_2$ during night time at 5 out of
7 stations, with means ($\pm$ SE) of $-54.6 \pm 37$ µmol $CO_2$ $m^{-2}$ $h^{-1}$ and $86 \pm 120$ µmol $CO_2$ $m^{-2}$ $h^{-1}$
respectively (Table 1, Fig. 2). However, in three out of seven sites, heterotrophic activities
outbalanced photosynthesis on a 24h basis. At two sites, S3 and S6, we found an increase of the
$CO_2$ emissions between day and night, contradictory to the classical daytime primary
production–night-time respiration pattern, possibly indicative of a light mediated increase of
heterotrophic processes.

Methane emissions did not show circadian patterns with linear increases in $CH_4$ concentration in
our incubations (Fig. 2) and with similar light and dark rates ($0.26 \pm 0.08$ and $0.21 \pm 0.07$ µmol
$CH_4$ $m^{-2}$ $h^{-1}$ (mean $\pm$ SE), respectively (Table 1). In terms of total GHG contribution, the mean
$CO_2$-equivalents ($CO_2e$) emission to the atmosphere was $564 \pm 1284$ µmol $CO_2e$ $m^{-2}$ $d^{-1}$ (mean $\pm$
SE) using the 100 years' time horizon global warming potential (Myhre et al., 2013). Inundated
mangrove sediments were net emitters of $CO_2e$ in three out of seven sites (Table 1), and in five
out of seven mangrove stands sampled, $CH_4$ was the main source of $CO_2e$ to the atmosphere.

While no overall trend was revealed through the relationship between day and night fluxes for
$CO_2$ and $CH_4$ (Fig. 3), consistencies are evident at specific mangrove stations. For example,
night $CO_2$ emissions are clearly visible at S2, while S3 appears to emit $CO_2$ during daylight
hours. No relationship was apparent between GHG fluxes and the densities of organic carbon or
nitrogen in the sediment. There was no discernible trend between gas fluxes and chlorophyll *a*
content in surface sediments.

The isotopic signatures of the produced $CO_2$ ($\delta^{13}C$-$CO_2$) ranged from -11.21 to -25.72 ‰ as
derived from keeling plots (Fig. 4, Table 1). The $\delta^{13}C$-$CO_2$ was similar for almost all stations,
with the exception of S3 that had a $\delta^{13}C$-$CO_2$ of -25.72 ‰. The isotopic composition of the
produced $CH_4$ ($\delta^{13}C$-$CH_4$) ranged from -71.28 to -87.08 ‰, with a mean $\delta^{13}C$ signature of -80.61
‰ (Fig. 4, Table 1).

**4      Discussion**

**4.1    Greenhouse gas fluxes**

The $CO_2$ and $CH_4$ emissions reported in this study show that Red Sea mangroves can act as a
source of GHG to the atmosphere. Values reported from this study fall within previously
reported ranges for both $CH_4$ and $CO_2$, but maximum $CH_4$ and $CO_2$ flux rates in the Red Sea are
up to 100 fold below those reported elsewhere. Compiled global values for GHG fluxes range
from -16.9 to 629.2 mmol $CO_2$ m$^{-2}$ d$^{-1}$ and -2.1 to 25,974 µmol $CH_4$ m$^{-2}$ d$^{-1}$, with mean (±SE)
maximum emission rates averaging 202.3 ± 48 mmol m$^{-2}$ d$^{-1}$ and 4783.6 ± 2783 µmol m$^{-2}$ d$^{-1}$ for
$CO_2$ and $CH_4$ respectively (Table 2).

The variability in GHG emission rates reported in this study could be attributed to spatial
differences, as cores were taken from different parts of each forest. Indeed, previous studies
report significant discrepancies in emission rates in fringe versus forest positions (Allen et al.,
2007). Additionally it is possible that differences in flux rates may exist as a result of sediment
disturbance from the coring process. The effects of mangrove pneumatophores and possible
bioturbation from infaunal species such as burrowing crabs were not considered here yet could
pose another possible source of variation in results as the presence of these structures influences
oxygenation of sediment and pore water exchange, identified as driving factors in varying $CO_2$
levels (Call et al., 2014; Rosentreter et al., 2018). These factors likely affect relevant redox
processes and would therefore be useful to quantify in future studies.

Uniformity of day and night emission rates for $CH_4$ was observed in Red Sea mangrove stands,
with mean ($\pm$ SE) $CH_4$ emission rates of $0.28 \pm 0.08$ µmol $CH_4$ $m^{-2}$ $h^{-1}$ during the day and $0.24 \pm$
$0.08$ µmol $CH_4$ $m^{-2}$ $h^{-1}$ during night; this is consistent with previous work reporting that emission
rates for $CH_4$ do not vary significantly during light and dark hours in mangrove forests (Allen et
al., 2007). It has been suggested instead that variables such as sediment temperature are more
significant in their contributions to emission rates (Allen et al., 2007; Allen et al., 2011).
Incubated sediment cores kept at constant temperature do not reflect the range of temperatures
experienced by mangrove sediments over the diurnal cycle; future studies examining GHG
emissions under more realistic temperature fluctuations are needed. Seasonal studies of longer
duration have reported increased emission rates during warmer seasons (Chen et al., 2016;
Livesley and Andrusiak, 2012). Methane concentrations typically remain low due to anaerobic
methane oxidation processes that take place near sediment surfaces (Kristensen et al., 2008a),
consistent with the low $CH_4$ emission rates from Red Sea mangrove sediments observed here.
Additionally, environments of high salinity like the Red Sea have been associated with decreased
$CH_4$ emissions, as sulfate-reducing bacteria are thought to outcompete methanogens
(Poffenbarger et al., 2011).

Methane emission rates at the air-sea interface of Red Sea mangrove sediments, although quite
low, become more substantial when considered in terms of global warming potential. In this
study, $CH_4$ was, despite the comparatively low emission rates, the main source of $CO_2e$ in the
majority of sampled mangroves, and therefore deserves careful monitoring in this region.
Reported organic carbon burial rates of Red Sea mangroves of $3.42$ mmol C $m^{-2}$ $d^{-1}$
(Almahasheer et al. 2017) are 10 times larger than the combined average $CO_2$ and $CH_4$ emission
rates reported here ($0.37$ mmol C $m^{-2}$ $d^{-1}$), suggesting that these mangrove sediments could act as
net atmospheric carbon sinks; however, significant alkalinity and DIC exports have been
identified from mangroves as well (Sippo et al., 2016), necessitating future studies which
measure these exports to neighboring habitats in order to close the carbon budget and determine
the role of Red Sea mangroves in potential climate change mitigation. Currently, protection
measures and further reforestation efforts are being deployed along the Red Sea that will further
expand the area of mangroves (Almhasheer et al. 2016). The rationale for conserving mangroves
in the climate change context is not adequately represented by their net carbon sink capacity
when undisturbed, but rather by the emissions resulting from their disturbance. Indeed, previous
studies analyzing anthropogenic impacts on methane emission rates from mangrove sediments
have shown that disturbance significantly increases methane emissions (Purvaja and Ramesh,
2001; Chen et al., 2011). This provides an additional rationale to conserve, and continue to
expand, Red Sea mangroves.

While this study provides new insights on GHG fluxes from arid mangroves, the methods used
here solely measure the air-sea fluxes of dissolved gases. If $CO_2$ is produced from underlying
sediments, it enters the water column and becomes a part of the carbonate system, with
possibility of conversion to bicarbonate ($HCO_3^-$) and carbonate ($CO_3^{2-}$) ions; these dominating
species represent over 99% of the dissolved inorganic carbon (DIC) under current atmospheric
and oceanic conditions (Zeebe and Wolf-Gladrow, 2001). Therefore, the air-sea equilibration
methods used in this study do not measure DIC fluxes, but only the fluxes of the dissolved $CO_{2^-}$
component of this larger system.

Frankignoulle and Borges (2001) show that $CO_2$ can be measured either directly (using
equilibrator techniques and spectroscopy or chromatography) or indirectly (by making
calculations based on pH, total alkalinity, and DIC). The methodology presented in this study
represents the former, utilizing an air-sea equilibrator connected to a CRDS to measure GHG
fluxes at the air-sea interface. Research conducted by Borges et al. (2003) utilizes the indirect
approach, using pH and total alkalinity measurements in Papua New Guinea to calculate DIC
and $CO_{2(dis)}$ (for a computational discussion see Frankignoulle and Borges, 2001). Both methods
measure at the air-sea interface (Table Two) but are not directly comparable, as a full
determination of the carbonate system was not carried out in the present study. Similarly, studies
using equilibrator techniques that measure the dissolved $CO_2$ fraction of seawater to the
atmosphere are influenced by the seawater carbonate system and further steps of isotopic fraction
(discussed below), and are therefore not directly comparable to those studies which measure
GHG fluxes from exposed mangrove sediments to the atmosphere (Table Two).

**4.2    Isotopic composition of emitted gases**

There were no relationships between GHG fluxes and sediment properties, such as chlorophyll *a*,
nitrogen density, and organic carbon density, suggesting that other factors have greater influence
on GHG flux rates in this region. Since mangroves can receive large contributions of organic
carbon from other sources (Newell et al., 1995), such as algal mats, seagrass and seaweed,
examination of the isotopic composition of emitted carbon provides insights into the origin of the
organic carbon supporting GHG fluxes in mangrove sediments; however, it should be noted that
$\delta^{13}C$ values reported in this study occur after several steps of isotopic fractionation and may
therefore influence results. Isotope effects can cause an unequal distribution of isotopes between
DIC components; for example as $CO_2$ is produced from mangrove sediments and becomes part
of the carbonate system (likely forming $HCO_3^-$ after equilibration), molecules containing the
heavier carbon isotope—with a higher activation energy—will typically react more slowly
(Zeebe and Wolf-Gladrow, 2001), promoting a higher concentration of the heavy isotope in
unreacted $CO_2$ and a relative depletion of this heavier isotope in resulting $HCO_3^-$. Similarly, this
preferential incorporation and movement of molecules containing lighter isotopes can affect
resulting carbon isotope ratios after air-sea equilibration (with depletion of lighter isotopes in
seawater as a result of fractionation). $CO_2$ measured in this study is subject to these processes
and may not reflect the isotopic ratios of carbon originally emitted; rather, the signatures
measured in this study should be seen as a proxy which reflects isotopic ratios of air-sea
discrimination and biological processing (decomposition, respiration, and photosynthesis),
resulting after carbon isotope fractionation. Interpretation of results is therefore subject to this
limitation.

The isotopic signature of the $CO_2$ ($\delta^{13}C$-$CO_2$) produced by mangrove sediments in four out of the
five mangrove stands with available isotopic data was heavier (from -11.2 ± 0.6 to -15.9 ± 1.1
‰; Table 1) than the isotopic signature of mangrove tissues, suggesting decomposition of
organic matter from mixed sources (Kennedy et al. 2010). Specifically, the isotopic signature of
the mangroves found in the central Red Sea has been recently reported as $\delta^{13}C_{leaves}$ = -26.98 ±
0.15 ‰, $\delta^{13}C_{stems}$ = -25.75 ± 0.16 ‰ and $\delta^{13}C_{roots}$ = -24.90 ± 0.17 ‰ for mangrove leaves, stems
and roots while the mean isotopic signature of other primary producers in the central the Red Sea
has been reported as $\delta^{13}C_{seaweed}$ = -12.8 ± 0.5 ‰ and $\delta^{13}C_{seagrass}$ = -8.2 ± 0.2 ‰ for seaweed and
seagrass tissues, respectively (Almahasheer et al. 2017). However, in one mangrove stand (S3)
the $\delta^{13}C$-$CO_2$ was much lighter (-25.72 ± 0.21 ‰), potentially indicating mangrove tissues. Thus,
according to the isotopic signature, the $CO_2$ produced in mangrove sediments would be
supported by mangrove biomass in only one mangrove stand out of the five sampled sites with
available isotopic data. Moreover, the mean isotopic signature of the $CH_4$ produced in mangrove
sediments ($\delta^{13}C$-$CH_4$ = -80.6 ‰) tentatively confirms its biogenic origin, which normally ranges
from -40 to -80 ‰, depending on the isotopic signature of the organic compounds being
biologically decomposed (Reeburgh, 2014). The lowest $\delta^{13}C$-$CH_4$ was detected in S3, coinciding
with the lowest $\delta^{13}C$-$CO_2$ value, suggesting that the organic matter being decomposed by
methanogens likely came from mangrove tissues as well.

Interestingly, the mangrove with the lightest $\delta^{13}C$-$CO_2$ and $\delta^{13}C$-$CH_4$ (S3), showed the lowest
daily $CO_2$ flux (-1524 ± 686 µmol $CO_2$ m$^{-2}$ d$^{-1}$) but the highest $CH_4$ emission rate (13.3 ± 9.5
µmol $CH_4$ m$^{-2}$ d$^{-1}$), compared to the fluxes detected in the rest of mangrove stands with available
isotopic data. Part of the variability in the $CO_2$ ($R^2$ = 0.42) and $CH_4$ ($R^2$ = 0.40) emission rate
seems to be explained by the origin of the organic matter being decomposed, estimated here as
the $\delta^{13}C$-$CO_2$ and $\delta^{13}C$-$CH_4$. Organic matter with lighter isotopic composition could enhance
$CO_2$ emissions, whereas organic matter with heavier isotopic composition could enhance $CH_4$
emissions (Fig. 5), possibly suggesting a different preferential use of organic matter by different
microbial groups in mangrove sediments. Future studies exploring this idea with further
considerations of carbon isotope fractionation would help solidify the role of the origin of
organic carbon stored in mangrove sediments on their GHG emissions.

**5       Conclusion**

This study is first in reporting $CO_2$ and $CH_4$ fluxes from Red Sea mangrove sediments,
contributing to the scant data on arid mangrove systems (Atwood et al. 2017, Almahasheer et al.
2017), essential to establish a solid baseline on GHG emissions for future studies. Results show
that maximum $CO_2$ and $CH_4$ flux rates from Red Sea mangrove sediments are well below those
reported elsewhere, and that, even when considered in terms of $CO_2$ equivalents, carbon burial
rates largely outweigh GHG emission rates at the air-sea interface in this region. This study also
highlights the importance of determining the source of organic matter in GHG flux studies, as
emissions appear to be supported by organic matter from mixed sources in the majority of
studied mangroves, potentially enhancing $CH_4$ production over $CO_2$ fluxes in this system.
Seasonal variation should be considered in future studies on GHG emissions by Red Sea
mangroves to better determine annual emission rates from this system, which reaches some of
the warmest temperatures experienced by mangrove forests worldwide. Similarly, a wider spatial
coverage within the mangrove forest should be considered to confidently determine net GHG
fluxes that can be upscaled to the entire stock of Red Sea mangroves.
Methods presented in this study include the use of an air-sea equilibrator connected to a CRDS to
measure GHG fluxes at the air-sea interface, measuring the dissolved $CO_2$ component of the
larger seawater carbonate system. This methodology is one of many used to measure GHG flux
rates; establishing a unified sampling technique at both the soil-atmosphere and air-seawater
interface will aid future researchers in determining total carbon budgets and accurately informing
policymakers of their findings. In combination with consideration of isotope effects, a full
determination of the carbonate system will be beneficial in future studies to further resolve GHG
fluxes in arid mangroves, allowing us to better ascertain the role of these forests in global carbon
budgets.

*Data availability.* All data will be accessible in the repository Pangea pending manuscript
acceptance.

*Competing interests.* The authors declare that they have no conflict of interest.

**Author contribution**
MAS, NG-B, VS and CMD designed the study. MAS and NG-B performed the measurements
and calculations. MAS, NG-B, VS and CMD interpreted the results. All authors contributed
substantially to the final manuscript.

**Acknowledgements**

This research was funded by King Abdullah University of Science and Technology (KAUST)
through baseline funding to C.M.D.  We thank D. Krause-Jensen, N. Massoudi, and K. Baldry
for help during sampling, and the captain and crew of KAUST R/V Thuwal for support. M.A.S.
was supported by King Abdullah University of Science and Technology through the VRSP
program. We thank P. Carrillo de Albornoz for lab instrument support, and M. Ennasri for help
with sediment analysis.

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

FIGURE HEADINGS

**Fig. 1.** Mangrove stands sampled along the Saudi coast of the Red Sea. Numbers indicate
positions of sampling sites from this study. S1 and S2: King Abdullah University of Science and
Technology; S3: Duba; S4 and S5: Al Wahj; S6 and S7: Farasan Banks.

**Fig. 2.** Change in $CO_2$ (left panels) and $CH_4$ (right panels) concentrations over time in triplicated
mangrove sediment cores from mangrove stations S3-S7. Shaded areas represent night time and
each replicate is coded by different symbols.

**Fig. 3.** Relationship between day and night fluxes for $CO_2$ (top panel) and $CH_4$ (bottom panel) at
all mangrove stations.

**Fig. 4.** Keeling plots for mangrove stations S3-S7, showing the linear regression of the inverse of
$CO_2$ concentration (left panels) and $CH_4$ concentration (right panels) versus $\delta^{13}C$–$CO_2$ and $\delta^{13}C$–
$CH_4$. Y-intercepts were used to estimate the isotopic signatures of produced gases.

**Fig. 5.** Relation between the carbon isotopic signature of the produced $CO_2$ ($\delta^{13}C$–$CO_2$) and $CO_2$
fluxes (top panel) and carbon isotopic signature of the produced $CH_4$ ($\delta^{13}C$–$CH_4$) and the $CH_4$
fluxes (bottom panel) in Red Sea mangroves. Error bars indicate standard error of the mean.










**Figure 1**

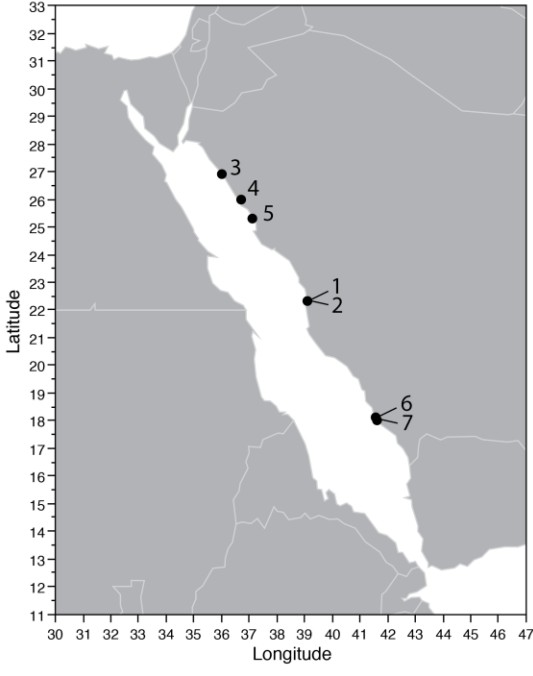










**Figure 2**

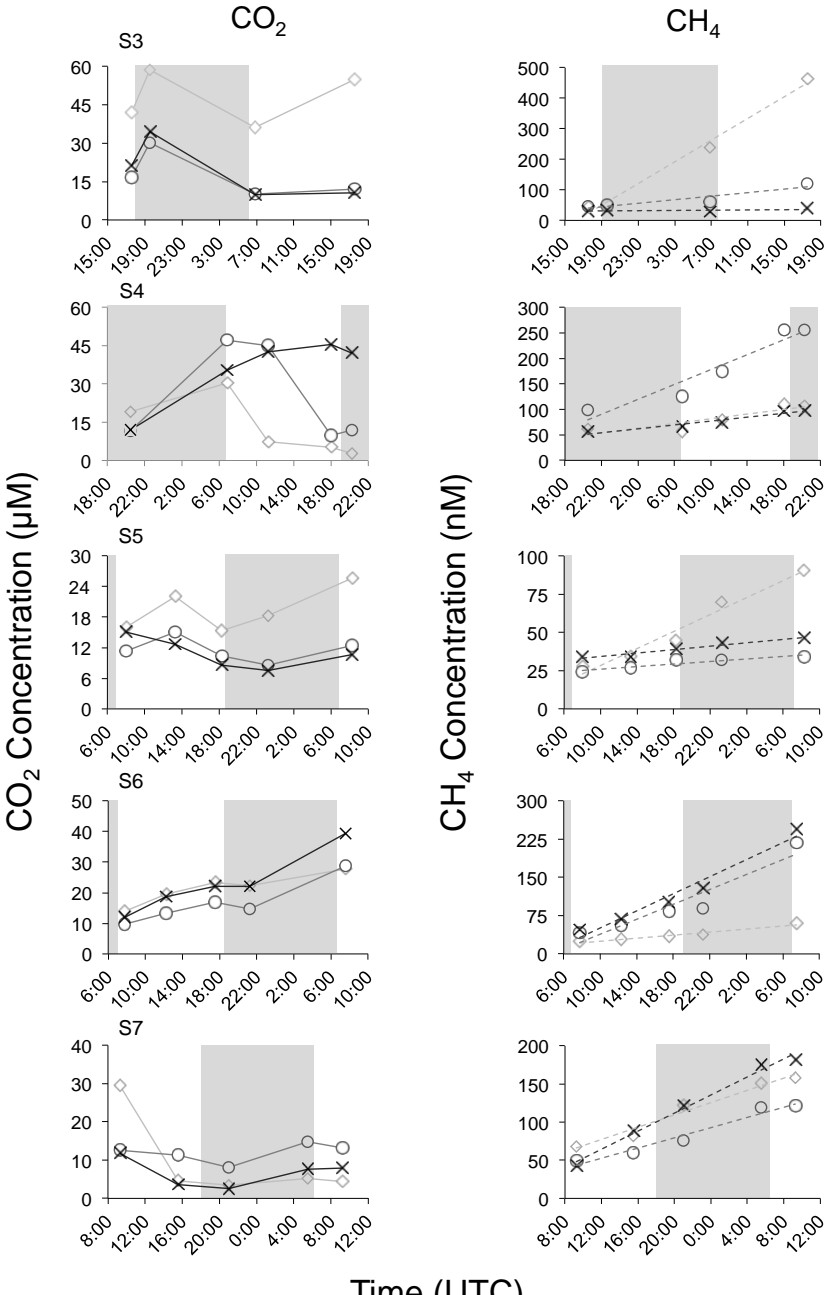




**Figure 3**

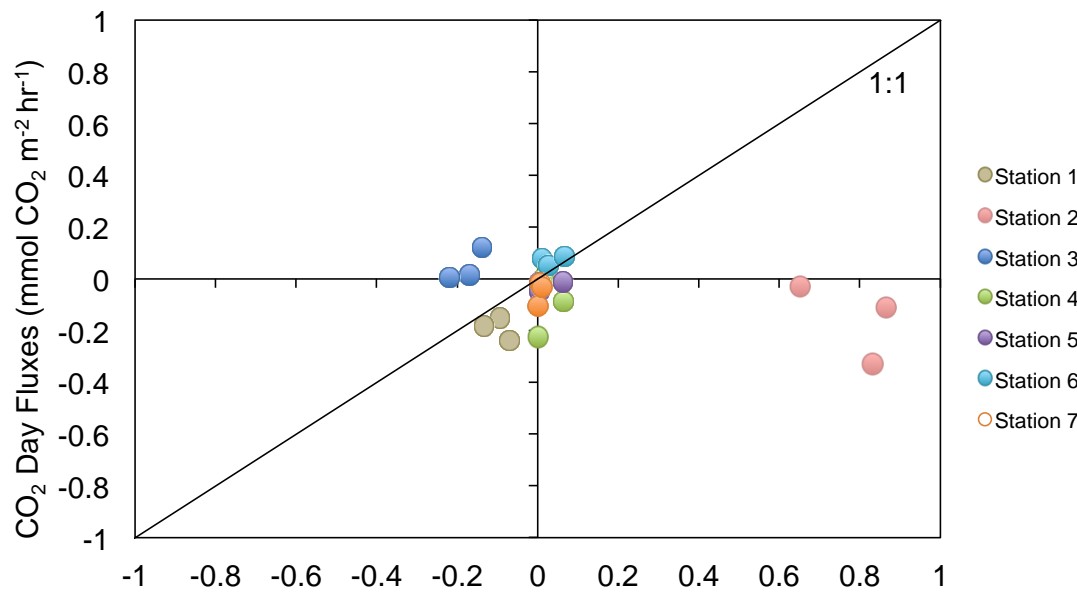

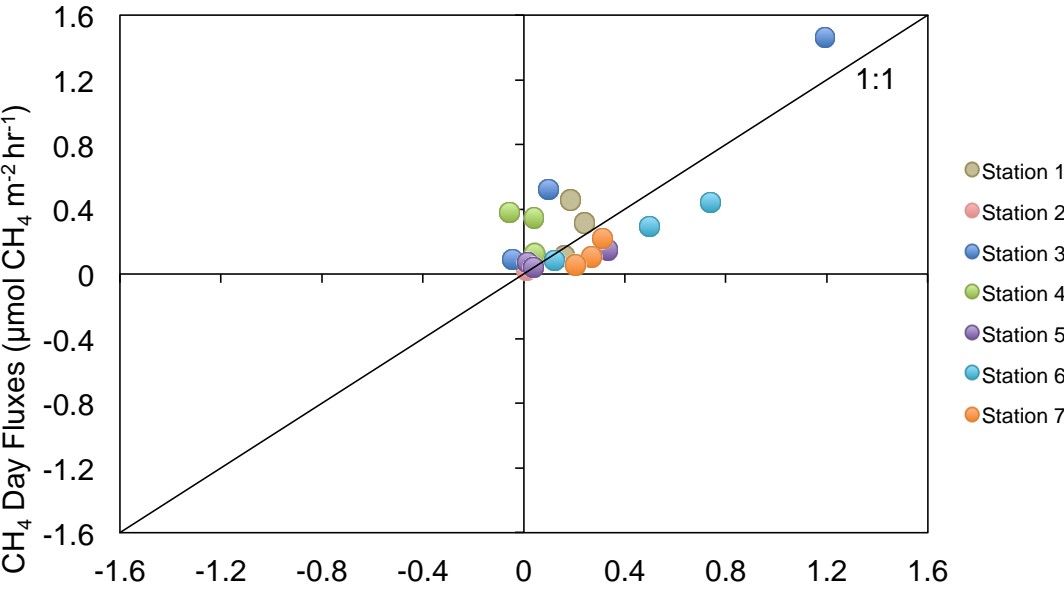

**Figure 4**

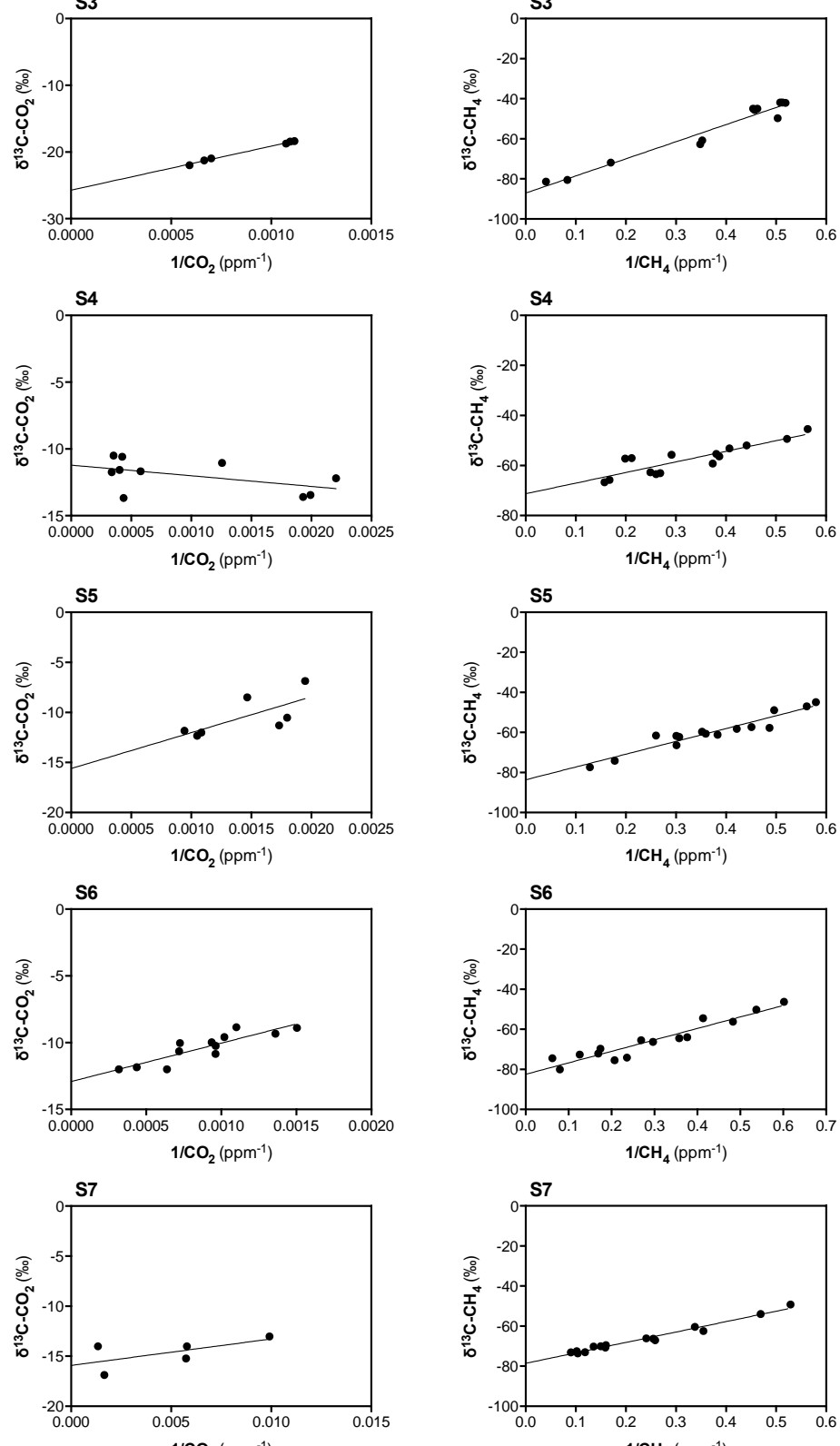

**Figure 5**


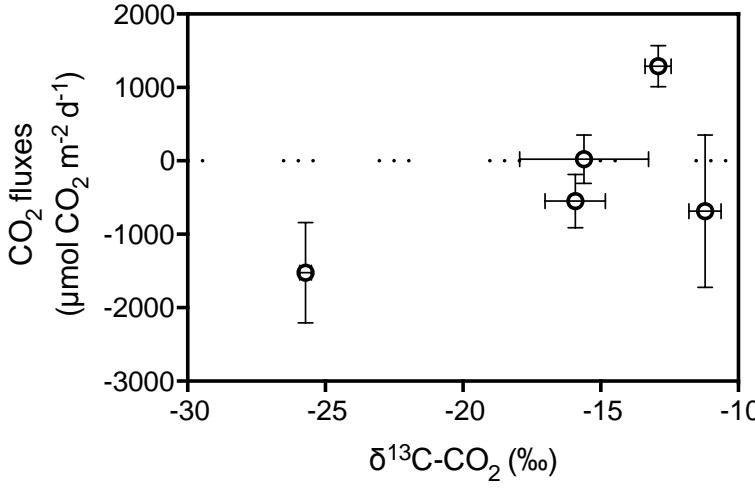

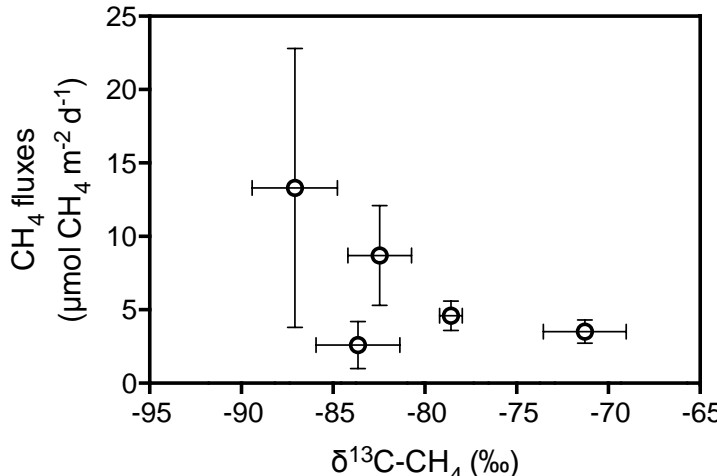


**Table 1.** Summary of greenhouse gas fluxes and sediment characteristics from studied mangrove forests. $CH_4$ fluxes in brackets represent $CO_2$ equivalents in terms of global warming potential for a time horizon of 100 years ($GWP_{100}$), taking into account climate-carbon feedback as suggested by the AR5 of IPCC (Myhre et al., 2013). Data represent the mean ± SEM and nd means no data available.

| Station | $CO_2$ Day Flux ($\mu$mol $CO_2$ m$^{-2}$ hr$^{-1}$) | $CH_4$ Day Flux ($\mu$mol $CH_4$ m$^{-2}$ hr$^{-1}$) | $CO_2$ Night Flux ($\mu$mol $CO_2$ m$^{-2}$ hr$^{-1}$) | $CH_4$ Night Flux ($\mu$mol $CH_4$ m$^{-2}$ hr$^{-1}$) | Daily $CO_2$ Flux ($\mu$mol $CO_2$ m$^{-2}$ d$^{-1}$) | Daily $CH_4$ Flux ($\mu$mol $CH_4$ m$^{-2}$ d$^{-1}$) | $\delta^{13}$C-$CO_2$ (‰) | $\delta^{13}$C-$CH_4$ (‰) | Nitrogen Density (mgN cm$^{-3}$) | $C_{org}$ Density (mgC cm$^{-3}$) | Chl $a$ ($\mu$g Chl $a$/gr sediment) |
|---|---|---|---|---|---|---|---|---|---|---|---|
| 1 | -188 ± 25 | 0.30 ± 0.17 [10.2] | -99 ± 18 | 0.19 ± 0.04 [6.46] | -3452 ± 271 | 5.9 ± 1.3 [201] | nd | nd | nd | nd | nd |
| 2 | -157 ± 89 | 0.05 ± 0.02 [1.7] | 782 ± 66 | 0.03 ± 0.01 [1.02] | 7500 ± 894 | 0.9 ± 0.25 [31] | nd | nd | nd | nd | nd |
| 3 | 49 ± 37 | 0.69 ± 0.4 [23.46] | -176 ± 23 | 0.42 ± 0.39 [14.28] | -1524 ± 686 | 13.3 ± 9.5 [452] | -25.7 ± 0.2 | -87.1 ± 2.3 | 1.03 ± 0.05 | 13.33 ± 1.01 | nd |
| 4 | -86 ± 79 | 0.28 ± 0.1 [9.52] | 29 ± 19 | 0.01 ± 0.03 [0.34] | -684 ± 1038 | 3.5 ± 0.8 [119] | -11.1 ± 0.6 | -71.3 ± 2.3 | 0.80 ± 0.03 | 8.98 ± 0.86 | 1.02 ± 0.05 |
| 5 | -22 ± 11 | 0.09 ± 0.03 [3.06] | 24 ± 20 | 0.13 ± 0.10 [4.42] | 23 ± 331 | 2.6 ± 1.6 [88] | -15.6 ± 2.3 | -83.6 ± 2.3 | 1.12 ± 0.05 | 13.34 ± 0.98 | 1.03 ± 0.04 |
| 6 | 73 ± 10 | 0.27 ± 0.10 [9.18] | 35 ± 17 | 0.45 ± 0.18 [15.30] | 1289 ± 280 | 8.7 ± 3.4 [296] | -12.9 ± 0.5 | -82.5 ± 1.7 | 1.51 ± 0.14 | 10.58 ± 0.82 | 0.43 ± 0.14 |
| 7 | -51 ± 28 | 0.13 ± 0.05 [4.42] | 5 ± 3 | 0.26 ± 0.03 [8.84] | -547 ± 363 | 4.6 ± 1.0 [156] | -15.9 ± 1.1 | -78.6 ± 0.6 | 3.30 ± 0.55 | 33.43 ± 6.69 | 1.86 ± 0.12 |

**Table 2.** Comparison of GHG fluxes from global mangrove forests and Red Sea mangroves. Literature values converted from reported form for comparison purposes. Measurements made at the: 1. soil-atmosphere interface, 2. air-sea interface with DIC calculation methods, and 3. air-sea interface with equilibration methods.

| | | | $CO_2$ (mmol m$^{-2}$ d$^{-1}$) | | $CH_4$ (µmol m$^{-2}$ d$^{-1}$) | |
|---|---|---|---|---|---|---|
| Author | Year | Place | Minimum | Maximum | Minimum | Maximum |
| Allen et al.[1] | 2007 | Australia | - | - | 4.5 | 25974 |
| Allen et al.[1] | 2011 | Australia | - | - | 70.3 | 2348 |
| Alongi et al.[1] | 2005 | China | 17 | 121 | 5 | 66 |
| Chen et al.[1] | 2016 | China | -16.9 | 279.2 | -2.1 | 8015.1 |
| Kristensen et al.[1,2] | 2008b | Tanzania | 28 | 115 | 0 | 87.6 |
| Livesley & Andrusiak[1] | 2012 | Australia | 50 | 150 | 50 | 749 |
| Borges et al.[2] | 2003 | Papua New Guinea | - | 43.6 | - | - |
| Bouillon et al.[2] | 2003 | India | - | 70.2 | - | - |
| Bouillon et al.[2] | 2007a | Kenya | 3 | 252 | - | - |
| Bouillon et al.[2] | 2007b | Kenya | - | 52 | - | - |
| Bouillon et al.[2] | 2007c | Tanzania | 1 | 80 | - | - |
| Call et al.[3] | 2015 | Australia | 9.4 | 629.2 | 13.1 | 632.9 |

| Ho et al. [3] | 2014 | United States | 20 | 118 | - | - |
|---|---|---|---|---|---|---|
| Jacotot et al.[3] | 2018 | New Caledonia | 3.12 | 441.8 | 4.32 | 4129.7 |
| Rosentreter et al.[3] | 2018a | Australia | 58.7 | 277.6 | - | - |
| Rosentreter et al.[3] | 2018b | Australia | - | - | 96.5 | 1049.8 |
| This Study[3] | 2017 | Red Sea | -3.5 | 7.5 | 0.9 | 13.3 |