# Peer review of "Carbon Dioxide and Methane Fluxes at the Air-Sea Interface of Red"

_Biogeosciences, 2018_

## Referee Comment (RC1) · Anonymous Referee #1 · 15 Feb 2018

Sea et al. report data on CO2 and CH4 fluxes from a series of mangrove sediments along the Saudi Arabian coast, as well as data on d13C of CO2 and CH4 from their incubation experiments. While the topic is certainly of interest given the relative scarcity of GHG flux measurements from mangrove sediments in arid zones, unfortunately I disagree with the methodology and experimental appraoch, which in my opinion renders the CO2 and d13C-CO2 data incorrect. As outlined below, the methodology does not account for the inorganic carbon system equilibrium (CO2 produced will rapidly re-equilibrate with bicarbonate and carbonate ions) and for isotope fractionation between the different inorganic C species.

Detailed comments

I focus here only on the methodology aspects since in my opinion the approach does

not measure CO2 emission rates correctly, and the d13C data are similarly not representative. These concerns invalidate the discussion and conclusions on CO2 fluxes and sources of CO2. The authors used 2 different experimental setups to perform their incubations. In both cases, CO2 fluxes are derived from an increase in the partial pressure of CO2 in the water column overlying the sediment in a sediment core; either by directly measuring pCO2 in the water (described in section 2.3.1) or by measuring pCO2 in the headspace above the water column in a sediment-water-headspace incubation (described in section 2.3.2). Both approaches assume that all CO2 produced in – and released from – the sediment accumulates as CO2 in the overlying water, and equilibrates with CO2 in the headspace (for the 2nd approach), but ignores the fact that dissolved CO2 will rapidly equilibrate with dissolved HCO3- and CO3—(see for example Schulz et al. Marine Chemistry 100: 53-65 for a discussion on the kinetics of the inorganic carbon equilibration). What should be determined is the change in the total DIC concentration, rather than only looking at CO2. In addition, anaerobic minerization processes within the sediment (sulphate reduction is likely important) may result in the release of bicarbonate rather than CO2, further necessitating the use of total DIC concentration data. The same problem holds for the d13C data, which was measured on CO2 in the headspace (2nd appraoch). Since there is isotope fractionation in the inorganic C system, with CO2 being substantially depleted in 13C relative to bicarbonate, the changes in d13C in CO2 in the headspace are not directly linked to the CO2 produced in the sediment by respiration, but are transformed during equilibration in the water column overlying the sediment, and there is an additional fractionation step between aquous (dissolved) and gaseous CO2 (in the headspace). Hence, the Keeling plot approach will not provide a reliable way of determining the source of CO2 produced – both the concentration and the d13C data determined in the authors' approach are not relevant; it is the total DIC concentrationa and d13C of the total DIC pool (or rather, DIC + CO2 in headspace) that should have been measured.

The CH4 data do appear valid, since they do not suffer from the issues described above for CO2. In principle however, isotope data on methane in the headspace should also

be corrected for fractionation between CH4 in the water and gas phase; although this will have a marginal effect on the final data and interpretation.

---

## Referee Comment (RC2) · Anonymous Referee #2 · 11 Apr 2018

The manuscript "Carbon dioxide and methane emissions from Red Sea mangrove sediments" details the spatial and diel flux of $CO_2$ and $CH_4$ from mangrove sediments in the understudied Red Sea region. The study uses carbon isotopes to differentiate the source of the gasses from sources other than mangrove tissue alone. The manuscript is well written and the data closes a key knowledge gap in global mangrove carbon cycling by providing the first estimate of sediment fluxes for the Red Sea.

Minor comments are listed below.

Line 134 For cores S1 and S1, you need to factor in the equilibration time of the membrane equilibrator as this would affect your rate calculations (Webb 2016 L&O). By not accounting for equilibration time the flux estimates would underestimate emission rates.

Line 46 Should be 12%

Line 198 Using the data in Table 1, I calculate a mean CO2 flux of 1358 $\pm$ 1195 umol m-2 day-1

Line 201 You do not include the negative flux numbers in the reported range. I find the variability of the source/sink behaviour of CO2 at the different sites to be one of the most interesting findings of the paper and there is limited speculation or use of the literature to suggest why that may be. I would suggest a deeper interpretation is necessary. Factors including the disturbance if sediments during coring may be particularly relevant as crab burrows would no doubt be affected and coring through mangrove roots may disturb the entire sediment matrix.

Line 202 It was 5 out of the 7 sites where daytime uptake and night time production was seen.

Line 203 the units should be umol CO2 m-2 hr-1

Line 231 Averages and standard errors would be useful in Table 2

Line 231 Including a supplementary map of each field site would help delineate potential differences between the sites.

Line 263 Fix reference

---

## Author Comment (AC1) · 15 May 2018

We would like to thank both anonymous reviewers for their thoughts, which will greatly enhance the thoroughness and readability of our manuscript.

RC 1: "Both [methodological] approaches assume that all CO2 produced in – and released from – the sediment accumulates as CO2 in the overlying water, and equilibrates with CO2 in the headspace (for the 2nd approach), but ignores the fact that dissolved CO2 will rapidly equilibrate with dissolved HCO3- and CO32-... What should be determined is the change in the total DIC concentration, rather than only looking at CO2."

AC 1: Reviewer one provides thought-provoking insights on carbonate chemistry in

seawater, with specific concerns focusing on the need to consider DIC fractionation in order to properly assess the fluxes of DIC between sediment and water due to photosynthesis, respiration and numerous Red-Ox processes. We absolutely agree with the comment of reviewer 1, however, our intent is to measure the fluxes at the air-sea interface, i.e. quantify the net GHG emissions to the atmosphere. It seems that our language in several places made our intent more ambiguous than realized. Air-sea equilibrators have been used in numerous studies to assess net CO2 emissions by marine ecosystems, such as in Borges et al., 2003 in inundated mangrove swamps.

In a newer version of the MS, we will make our scientific goals and interests more transparent, notably by clarifying that our study does not measure the fluxes of carbon from the sediment as only a full determination of the CO2 system would allow to do so, but only the potential net emission of CO2 to the atmosphere from the biological activities in the sediment and overlying water.

We will discuss the limitations of our method compared to the DIC variation method and perform a careful comparison between our results and the results from published studies, depending on the method used.

RC 2: "Since there is isotope fractionation in the inorganic C system, with CO2 being substantially depleted in 13C relative to bicarbonate, the changes in d13C in CO2 in the headspace are not directly linked to the CO2 produced in the sediment by respiration, but are transformed during equilibration in the water column overlying the sediment, and there is an additional fractionation step between aquous (dissolved) and gaseous CO2 (in the headspace)."

AC 2: We will clearly discuss the limitation of our method, acknowledging that our resulting d13C signals in the air phase come after several steps of isotopic segregation.

---

## Author Comment (AC2) · 15 May 2018

We would like to thank both anonymous reviewers for their thoughts, which will greatly enhance the thoroughness and readability of our manuscript.

RC 1: Line 134 For cores S1 and S1, you need to factor in the equilibration time of the membrane equilibrator as this would affect your rate calculations (Webb 2016 L&O). By not accounting for equilibration time the flux estimates would underestimate emission rates.

AC 1: We recognize that air-water equilibrators exhibit a delay in the measured response of gas concentration and that, for some applications requiring exact $CO_2$ concentrations at a given time, there is a need to deconvolve the $CO_2$ or $CH_4$ time series.

However, our study focused on rates, calculated as the slope during a phase in which we observed a linear increase or decrease of gases for periods of 20 to 30 minutes. Convolution of the time series due to lag would not affect those rates.

RC 2: Line 46 Should be 12%

AC 2: Line 46 will be changed from 13% to 12%.

RC 3: Line 198: Using the data in Table 1, I calculate a mean $CO_2$ flux of 1358 $\pm$ 1195 umol m-2 day-1

AC 3: It is possible that reviewer two calculated a mean $CO_2$ flux of 1358 umol m-2 day-1 if he or she accidentally plugged in +3452 for station 1 instead of -3452 for station 1. This would create a mean $CO_2$ flux of 1358 instead of 372.

RC 4: Line 201: You do not include the negative flux numbers in the reported range. I find the variability of the source/sink behaviour of $CO_2$ at the different sites to be one of the most interesting findings of the paper and there is limited speculation or use of the literature to suggest why that may be. I would suggest a deeper interpretation is necessary. Factors including the disturbance if sediments during coring may be particularly relevant as crab burrows would no doubt be affected and coring through mangrove roots may disturb the entire sediment matrix.

AC 4: We report the range of $CO_2$ fluxes observed to be -3452 to 7500 $\mu$mol $CO_2$ m-2 d-1; it is possible that reviewer two did not see the negative sign associated with -3452 as the negative symbol appears on line 200 while the number 3452 appears on line 201. We wholeheartedly agree with the reviewer's thoughts that the high degree of flux variability is an interesting finding and will subsequently add our thoughts on this in the discussion section of our manuscript.

RC 5: Line 202 It was 5 out of the 7 sites where daytime uptake and night time production was seen.

AC 5: Line 202 was originally written to denote an overall observation, as the majority of

sites absorbed CO2 during the day and emitted at night. We appreciate the reviewer's attention to detail and will change line 202 to "Mangrove sediments absorbed CO2 during daytime and emitted CO2 during night time at 5 out of 7 stations."

RC 6: Line 203 the units should be umol CO2 m-2 hr-1

AC 6: We apologize for this error; units will be corrected on line 203.

RC 7: Line 231 Averages and standard errors would be useful in Table 2

AC 7: In a next version of the MS we will provide range and mean ± SE.

RC 8: Line 231: Including a supplementary map of each field site would help delineate potential differences between the sites.

AC 8: While supplementary visuals would indeed aid in determining site differences, we unfortunately did not record exact core locations, but instead noted distance away from the forest edge, sampling near the center of the mangrove belt in each case. It was our hope that this would minimize spatial differences; regardless we felt the need to include the possibility of spatial variability in our discussion.

RC 9: Line 263: Fix reference

AC 9: As per the reviewer's suggestion, the referencing error on line 263 will be corrected.

---

## Author Response (AR1)

Mallory A. Sea
                                King Abdullah University of Science and Technology
                                                Red Sea Research Center (RSRC)
                                           Thuwal, Saudi Arabia, 23955-6900
                                                                   712-203-6729
                                                         mas012@morningside.edu

Prof. Michael Bahn
Co-editor-in-chief, *Biogeosciences*
Dear Prof. Michael Bahn,
Please find attached a revised version of the manuscript, **"Carbon Dioxide and Methane**
**Emissions from Red Sea Mangrove Sediments,"** now entitled **"Carbon Dioxide and**
**Methane Fluxes at the Air-Sea Interface of Red Sea Mangroves"** for publication
reconsideration in *Biogeosciences*.
We would like to sincerely thank the reviewers for their suggestions, which have significantly
improved the manuscript. We have attached a detailed response to the reviewers' comments,
noting all changes made to reflect their recommendations. We believe that these revisions have
greatly improved the manuscript's content and readability and hope you find this draft acceptable
for publication in its present form.
Yours sincerely,
Sea and co-authors

We wish to sincerely thank reviewer #1 for his/her comments and suggestions that were very helpful in improving and clarifying the MS.

Anonymous Referee #1:

Detailed comments

1. I focus here only on the methodology aspects since in my opinion the approach does not
measure CO2 emission rates correctly, and the d13C data are similarly not repre- sentative.
These concerns invalidate the discussion and conclusions on CO2 fluxes and sources of CO2.
The authors used 2 different experimental setups to perform their incubations. In both cases,
CO2 fluxes are derived from an increase in the partial pressure of CO2 in the water column
overlying the sediment in a sediment core; either by directly measuring pCO2 in the water
(described in section 2.3.1) or by measuring pCO2 in the headspace above the water column in a
sediment-water-headspace incu- bation (described in section 2.3.2). Both approaches assume that
all CO2 produced in – and released from – the sediment accumulates as CO2 in the overlying
water, and equilibrates with CO2 in the headspace (for the 2nd approach), but ignores the fact
that dissolved CO2 will rapidly equilibrate with dissolved HCO3- and CO32-(see for example
Schulz et al. Marine Chemistry 100: 53-65 for a discussion on the kinetics of the inorganic
carbon equilibration). What should be determined is the change in the total DIC concentration,
rather than only looking at CO2.

Reviewer one provides thought-provoking insights on carbonate chemistry in seawater, with
specific concerns focusing on the need to consider DIC fractionation in order to properly assess
the fluxes of DIC between sediment and water due to photosynthesis, respiration and numerous
Red-Ox processes.

We absolutely agree with the comment of reviewer 1, however, our intent is to measure the
fluxes at the air-sea interface, i.e. quantify the **net** GHG emissions to the atmosphere. It seems
that our language in several places made our intent more ambiguous than realized. Air-sea
equilibrators have been used in numerous studies to assess net $CO_2$ emissions by marine
ecosystems, such as in Borges et al., 2003 in inundated mangrove swamps.

In the new version of the MS, we make our scientific goals and interests more transparent
through numerous, related changes:

**Title**

The title has been changed to **"Carbon Dioxide and Methane Fluxes at the Air-Sea Interface**
**of Red Sea Mangroves"** to reflect this goal.

**Introduction**

We introduce the idea of measuring emissions in different ways in lines 74-78:
"Previous studies on GHG emission rates either focus on the soil-atmosphere interface,
highlighting substantial flux ranges with mangroves reported to act as negligible (Alongi, 2005)
to considerable sources (Livesley and Andrusiak, 2012; Chen et al., 2016), or examine net fluxes
at the air-sea interface, with few studies in arid systems."

We clarify our goals in lines 87-90:
"Here we report air-sea emission rates of $CO_2$ and $CH_4$, along with their carbon isotopic
composition, from incubations of inundated mangrove sediments cores along the Saudi coast of
the Red Sea. We assess the relative role of these two gases in supporting total GHG emissions as
well as their fluctuations along the day-night cycle."
**Discussion**
The discussion has now been divided into 2 subsections, the first of which addresses the
limitations of our methodology and subsequent comparison abilities (Lines 315-339):
*4.1 Greenhouse Gas Fluxes*
"While this study provides new insights on GHG fluxes from arid mangroves, the methods used
here solely measure the air-sea fluxes of dissolved gases. If $CO_2$ is produced from underlying
sediments, it enters the water column and becomes a part of the carbonate system, with
possibility of conversion to bicarbonate ($HCO_3^-$) and carbonate ($CO_3^{2-}$) ions; these dominating
species represent over 99% of the dissolved inorganic carbon (DIC) under current atmospheric
and oceanic conditions (Zeebe and Wolf-Gladrow, 2001). Therefore, the air-sea equilibration
methods used in this study do not measure DIC fluxes, but only the fluxes of the dissolved $CO_2$-
component of this larger system.
Frankignoulle and Borges (2001) show that $CO_2$ can be measured either directly (using
equilibrator techniques and spectroscopy or chromatography) or indirectly (by making
calculations based on pH, total alkalinity, and DIC). The methodology presented in this study
represents the former, utilizing an air-sea equilibrator connected to a CRDS to measure GHG
fluxes at the air-sea interface. Research conducted by Borges et al. (2003) utilizes the indirect
approach, using pH and total alkalinity measurements in Papua New Guinea to calculate DIC
and $CO_{2(dis)}$ (for a computational discussion see Frankignoulle and Borges, 2001). Both methods
measure at the air-sea interface (Table Two) but are not directly comparable, as a full
determination of the carbonate system was not carried out in the present study. Similarly, studies
using equilibrator techniques that measure the dissolved $CO_2$ fraction of seawater to the
atmosphere are influenced by the seawater carbonate system and further steps of isotopic fraction
(discussed below), and are therefore not directly comparable to those studies which measure
GHG fluxes from exposed mangrove sediments to the atmosphere (Table Two)."
We delete the possibility of closing the carbon budget with our study (lines 31-33 and 303-309)
and add the possibility of future studies to do so (lines 299-305):
"Reported organic carbon burial rates of Red Sea mangroves of 3.42 mmol C $m^{-2}$ $d^{-1}$
(Almahasheer et al. 2017) are 10 times larger than the combined average $CO_2$ and $CH_4$ emission
rates reported here (0.37 mmol C $m^{-2}$ $d^{-1}$), suggesting that these mangrove sediments could act as
net atmospheric carbon sinks; however, significant annual alkalinity and DIC exports have been
identified from mangroves as well (Sippo et al., 2016), necessitating future studies which
measure these exports to neighboring habitats in order to close the carbon budget and determine
the role of Red Sea mangroves in potential climate change mitigation."

**Conclusion**

Two concluding paragraphs have been added, tying together our thoughts and final remarks (lines 401-423). The second paragraph again addresses the limitations of our study (lines 415-423):

"Methods presented in this study include the use of an air-sea equilibrator connected to a CRDS to measure GHG fluxes at the air-sea interface, measuring the dissolved $CO_2$ component of the larger seawater carbonate system. This methodology is one of many used to measure GHG flux rates; establishing a unified sampling technique at both the soil-atmosphere and air-seawater interface will aid future researchers in determining total carbon budgets and accurately informing policymakers of their findings. In combination with consideration of isotope effects, a full determination of the carbonate system will be beneficial in future studies to further resolve GHG fluxes in arid mangroves, allowing us to better ascertain the role of these forests in global carbon budgets."

**Table Two**

References in Table Two have been added, deleted, and reorganized to better reflect related studies:

Deleted
Kristensen et al., 2008a
Alongi, 2014
Chuang et al., 2015

Added
Borges et al., 2003
Bouillon et al., 2003
Bouillon et al., 2007a
Bouillon et al., 2007b
Bouillon et al., 2007c
Call et al., 2015
Ho et al., 2014
Jacotot et al. 2018
Rosentreter et al. 2018a
Rosentreter et al. 2018b

Changes have also been made to Table Two to distinguish between measurements made at the [1] soil-atmosphere interface, [2] air-sea interface with DIC calculation methods, and [3] air-sea interface with equilibration methods.

In addition to these alterations, smaller word choice changes were made throughout the document to clarify our intent:
Line 77: "examine net fluxes at the air-sea interface"

Line 128: "We measured $CO_2$ and $CH_4$ air-sea fluxes"
Line 238-239: "Inundated mangrove sediments"
2. Since there is isotope fractionation in the inorganic C system, with CO2 being substantially
depleted in 13C relative to bicarbonate, the changes in d13C in CO2 in the headspace are not
directly linked to the CO2 produced in the sediment by respiration, but are transformed during
equilibration in the water column overlying the sediment, and there is an additional fractionation
step between aquous (dissolved) and gaseous CO2 (in the headspace). Hence, the Keeling plot
approach will not provide a reliable way of determining the source of CO2 produced – both the
concentration and the d13C data determined in the authors' approach are not relevant; it is the
total DIC concentrationa and d13C of the total DIC pool (or rather, DIC + CO2 in headspace)
that should have been measured.
We concur with Reviewer #1's  thoughts and consequently discuss the limitation of our method,
acknowledging that our resulting $d^{13}C$ signals in the air phase comes after several steps of
isotopic segregation. This is done in the following places:
**Abstract**
We make conditional conconclusions due to the limitations of our methods (Lines 24-29):
"Based on the isotopic composition of the $CO_2$ and $CH_4$ produced, we identified potential origins
of the organic matter that support GHG emissions. In all but one mangrove stand, GHG
emissions appear to be supported by organic matter from mixed sources, potentially reducing
$CO_2$ fluxes and instead enhancing $CH_4$ production, a finding that highlights the importance of
determining the origin of organic matter in GHG emissions"
**Discussion**
The discussion has now been divided into 2 subsections, the second of which addresses isotopic
segregation and its influence on our results (Lines 343-363):
*4.2 Isotopic Composition of Emitted Gases*
"There were no relationships between GHG fluxes and sediment properties, such as chlorophyll
*a*, nitrogen density, and organic carbon density, suggesting that other factors have greater
influence on GHG flux rates in this region. Since mangroves can receive large contributions of
organic carbon from other sources (Newell et al., 1995), such as algal mats, seagrass and
seaweed, examination of the isotopic composition of emitted carbon provides insights into the
origin of the organic carbon supporting GHG fluxes in mangrove sediments; however, it should
be noted that $\delta^{13}C$ values reported in this study occur after several steps of isotopic fractionation
and may therefore influence results. Isotope effects can cause an unequal distribution of isotopes
between DIC components; for example as $CO_2$ is produced from mangrove sediments and
becomes part of the carbonate system (likely forming $HCO_3^-$ after equilibration), molecules
containing the heavier carbon isotope—with a higher activation energy—will typically react
more slowly (Zeebe and Wolf-Gladrow, 2001), promoting a higher concentration of the heavy
isotope in unreacted $CO_2$ and a relative depletion of this heavier isotope in resulting $HCO_3^-$.

Similarly, this preferential incorporation and movement of molecules containing lighter isotopes can affect resulting carbon isotope ratios after air-sea equilibration (with depletion of lighter isotopes in seawater as a result of fractionation). $CO_2$ measured in this study is subject to these processes and may not reflect the isotopic ratios of carbon originally emitted; rather, the signatures measured in this study should be seen as a proxy which reflects isotopic ratios of air-sea discrimination and biological processing (decomposition, respiration, and photosynthesis) resulting after carbon isotope fractionation. Interpretation of results is therefore subject to this limitation."

We also make conditional conconclusions in this section due to the limitations of our methods :
Lines 380-385 : "Moreover, the mean isotopic signature of the $CH_4$ produced in mangrove sediments ($\delta^{13}$C-$CH_4$ = -80.6 ‰) tentatively confirms its biogenic origin"
Lines 387-389 : "The lowest $\delta^{13}$C-$CH_4$ was detected in S3, coinciding with the lowest $\delta^{13}$C-$CO_2$ value, suggesting that the organic matter being decomposed by methanogens likely came from mangrove tissues as well"
Lines 392-397: "Organic matter with lighter isotopic composition could enhance $CO_2$ emissions, whereas organic matter with heavier isotopic composition could enhance $CH_4$ emissions (Fig. 5), possibly suggesting a different preferential use of organic matter by different microbial groups in mangrove sediments. Future studies exploring this idea with further considerations of carbon isotope fractionation would help solidify the role of the origin of organic carbon stored in mangrove sediments on their GHG emissions."
Lines 406-409 : "This study also highlights the importance of determining the source of organic matter in GHG flux studies, as emissions appear to be supported by organic matter from mixed sources in the majority of studied mangroves, potentially enhancing $CH_4$ production over $CO_2$ fluxes in this system."

We would also like to thank reviewer #2 for his/her comments and suggestions that were very helpful in improving and clarifying the MS.

Anonymous Referee #2:

1. Line 134 For cores S1 and S1, you need to factor in the equilibration time of the membrane equilibrator as this would affect your rate calculations (Webb 2016 L&O). By not accounting for equilibration time the flux estimates would underestimate emission rates.

We recognize that air-water equilibrators exhibit a delay in the measured response of gas
concentration and that, for some applications requiring exact $CO_2$ concentrations at a given time,
there is a need to deconvolve the $CO_2$ or $CH_4$ time series.
However, our study focused on rates, calculated as the slope during a phase in which we
observed a linear increase or decrease of gases for periods of 20 to 30 minutes. Convolution of
the time series due to lag would not affect those rates.

2. Line 46 Should be 12%
Line 46 (now line 49) was changed from 13% to 12%.

3. Line 198: Using the data in Table 1, I calculate a mean CO2 flux of 1358 ± 1195 umol m-2
day-1
It is possible that reviewer two calculated a mean $CO_2$ flux of 1358 umol m-2 day-1 if he or she
accidentally plugged in +3452 for station 1 instead of -3452 for station 1. This would create a
mean CO2 flux of 1358 instead of 372.

4. Line 201: You do not include the negative flux numbers in the reported range. I find the
variability of the source/sink behaviour of CO2 at the different sites to be one of the most
interesting findings of the paper and there is limited speculation or use of the literature to suggest
why that may be. I would suggest a deeper interpretation is necessary. Factors including the
disturbance if sediments during coring may be particularly relevant as crab burrows would no
doubt be affected and coring through mangrove roots may disturb the entire sediment matrix.
We report the range of $CO_2$ fluxes observed to be -3452 to 7500 μmol CO2 m-2 d-1; it is
possible that reviewer two did not see the negative sign associated with -3452 as the negative
symbol appears on line 200 while the number 3452 appears on line 201. We moved the negative
sign down to the next line to make this more clear (now line 224). We wholeheartedly agree with
the reviewer's thoughts that the high degree of flux variability is an interesting finding and have
subsequently added our thoughts on this matter:

Lines 270-276: "Additionally it is possible that differences in flux rates may exist as a result of
sediment disturbance from the coring process. The effects of mangrove pneumatophores and
possible bioturbation from infaunal species such as burrowing crabs were not considered here yet
could pose another possible source of variation in results as the presence of these structures
influences oxygenation of sediment and pore water exchange, identified as driving factors in
varying $CO_2$ levels (Call et al., 2014; Rosentreter et al., 2018). These factors likely affect
relevant redox processes and would therefore be useful to quantify in future studies."

5. Line 202 It was 5 out of the 7 sites where daytime uptake and night time production was seen.
Line 202 was originally written to denote an overall observation, as the majority of sites
absorbed $CO_2$ during the day and emitted at night. We appreciate the reviewer's attention to
detail and have changed line 202 (now 226-227) to "Mangrove sediments absorbed $CO_2$ during
daytime and emitted $CO_2$ during night time at 5 out of 7 stations."

6. Line 203 the units should be umol CO2 m-2 hr-1
We apologize for this error; units were corrected on line 203 (now 227).

7. Line 231 Averages and standard errors would be useful in Table 2
As noted above, we feel that one of the more interesting findings in our study (and similar
studies) is the wide variability in reported flux values. We choose to keep flux ranges in Table 2
and add averages ($\pm$SE) in the text as suggested (lines 260-265):
Values reported from this study fall within previously reported ranges for both $CH_4$ and $CO_2$, but
maximum $CH_4$ and $CO_2$ flux rates in the Red Sea are up to 100 fold below those reported
elsewhere. Compiled global values for GHG fluxes range from -16.9 to 629.2 mmol $CO_2$ m$^{-2}$ d$^{-1}$
and -2.1 to 25,974 $\mu$mol $CH_4$ m$^{-2}$ d$^{-1}$, with mean ($\pm$SE) maximum emission rates averaging 202.3
$\pm$ 48 mmol m$^{-2}$ d$^{-1}$ and 4783.6 $\pm$ 2783 $\mu$mol m$^{-2}$ d$^{-1}$ for $CO_2$ and $CH_4$ respectively (Table 2).
8. Line 231: Including a supplementary map of each field site would help delineate potential
differences between the sites.
While supplementary visuals would indeed aid in determining site differences, we unfortunately
did not record exact core locations, but instead noted distance away from the forest edge,
sampling near the center of the mangrove belt in each case. It was our hope that this would
minimize spatial differences; regardless we felt the need to include the possibility of spatial
variability in our discussion.
9. Line 263: Fix reference
As per the reviewer's suggestion, the referencing error was corrected (now line 376).

[revised manuscript text omitted]